# IMProv: Inpainting-based Multimodal Prompting for Computer Vision Tasks

**Jiarui Xu**                                                    *jix026@ucsd.edu*
*UC San Diego*

**Yossi Gandelsman**                                     *yossi@gandelsman.com*
*UC Berkeley*

**Amir Bar**                                                 *amir.bar@berkeley.edu*
*UC Berkeley*

**Jianwei Yang**                                    *jianwei.yang@microsoft.com*
*Microsoft Research*

**Jianfeng Gao**                                            *jfgao@microsoft.com*
*Microsoft Research*

**Trevor Darrell**                                   *trevordarrell@berkeley.edu*
*UC Berkeley*

**Xiaolong Wang**                                              *xiw012@ucsd.edu*
*UC San Diego*

**Reviewed on OpenReview:** *https://openreview.net/forum?id=qBTgnk2HAf*

## Abstract

In-context learning allows adapting a model to new tasks given a task description at test time. In this paper, we present IMProv - a generative model that is able to in-context learn visual tasks from multimodal prompts. Given a textual description of a visual task (e.g. "Left: input image, Right: foreground segmentation"), a few input-output visual examples, or both, the model in-context learns to solve it for a new test input. We train a masked generative transformer on a new dataset of figures from computer vision papers and their associated captions, together with a captioned large-scale image-text dataset. During inference time, we prompt the model with text and/or image task example(s) and have the model inpaint the corresponding output. We show that training our model with text conditioning and scaling the dataset size improves in-context learning for computer vision tasks by over +10% AP for Foreground Segmentation, over +5% gains in AP for Single Object Detection, and almost 20% lower LPIPS in Colorization. Our emperical results suggest that vision and language prompts are complementary and it is advantageous to use both to achieve better in-context learning performance.

## 1  Introduction

In-context learning (ICL) (Brown et al., 2020; Chan et al., 2022; Xie et al., 2021), also known as few-shot prompting, is an exciting new paradigm in machine learning that allows a model to adapt to novel downstream tasks without fine-tuning or changing the model's weights. In natural language processing (NLP), ICL is considered an emergent property of large language models (Brown et al., 2020; Touvron et al., 2023; Chowdhery et al., 2022) and it was first introduced in the seminal paper of GPT-3 (Brown et al., 2020). A few-shot prompt typically includes examples of (input, output) pair(s). The few-shot performance of large

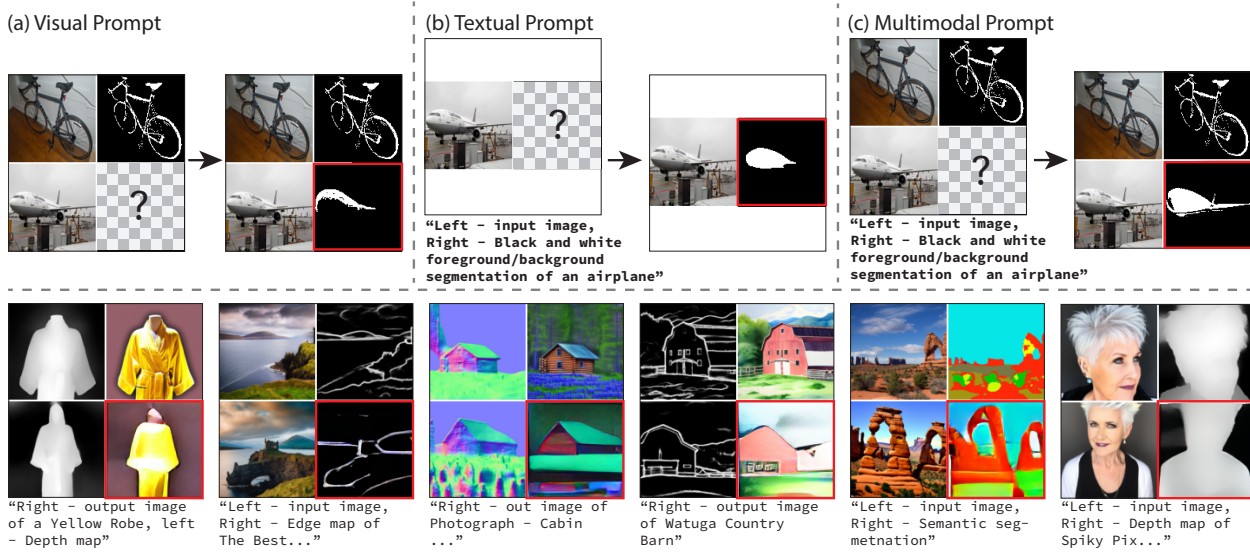

Figure 1: **I**npainting-based **M**ultimodal **Pro**mpting for **v**ision (IMProv). *Top:* Our model in-context learns to solve computer vision tasks by inpainting the masked area with the task solution (shown in red square) using visual input-output examples (a), a textual task description (b), or both (c). *Bottom:* IMProv prediction examples.

language models has been shown to achieve competitive results on NLP tasks, sometimes surpassing prior state-of-the-art fine-tuning approaches (Brown et al., 2020).

In computer vision, the full potential of in-context learning (ICL) is still far from being realized. To enable a model to perform in-context learning during test time, there are two key challenges that need to be addressed. Firstly, the model's architecture should be designed in such a way that it can effectively process prompts from various vision tasks. This means it should be capable of receiving task instructions and/or input-output examples as inputs to the model. Secondly, a different approach to training these models is required. While in natural language processing (NLP), the emergence of ICL has been facilitated by utilizing large-scale, non-curated data, in computer vision, even generative models trained on billions of non-curated text-image pairs have failed to achieve similar results.

A possible approach to enable test-time few-shot prompting for computer vision tasks is to train a multi-task inpainting model (Wang et al., 2022; 2023b; Bar et al., 2022). For example, previous approaches (Wang et al., 2022; 2023b) adopted a fully supervised paradigm to train a model over a predetermined set of vision tasks. However, this line of study requires a handful of manual annotations and thus struggles to scale and generalize well to unseen vision tasks. Instead of explicitly designing the tasks, Bar et al. (2022) took a different unsupervised approach by proposing to learn from unstructured Computer Vision Figures data, where images have implicit task supervision and grid-like structure. However, using vision-only prompting (Bar et al., 2022) suffers from ambiguities and is limited in its ability to describe a specific visual task.

To alleviate these difficulties, we propose to multimodal ICL by prompting using input that consists of both pixels and text. Intuitively, these two modalities can work in synergy to enhance the understanding of the world and its complexities. For example, during a conversation, people use language to communicate ideas and vision to perceive facial expressions and conversational gestures (Cassell et al., 1999; McNeill, 2019). For prompting, conditioning vision models on text can enable describing instructions in an efficient manner and reduce ambiguities without the necessity for multiple high-quality visual examples.

Equipped with this intuition, we train a model, dubbed IMProv, to inpaint randomly masked regions given the rest of the image and a caption as context *Our training does not require explicit definitions of tasks and annotations for each task.* To demonstrate multimodal learning can be boosted by larger dataset, we collected a new dataset of image-text pairs from Semantic Scholar, which is three times larger than the largest existing computer vision figures dataset. We train a new model by performing inpainting on randomly masked images from a combination of the newly constructed data and LAION 400M (Schuhmann et al., 2021).

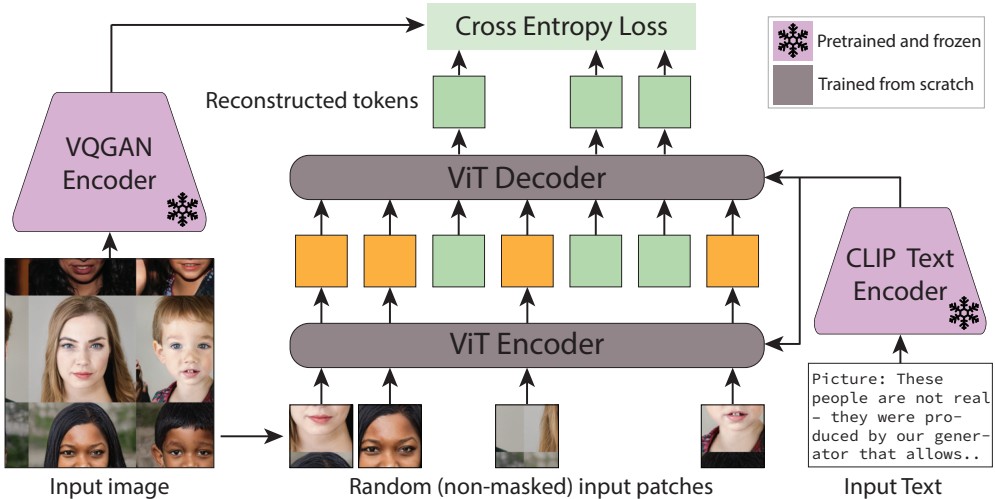

Figure 2: **IMProv Architecture**. During training, the input image is patchified, masked, and fed to the model together with the associated caption CLIP (Radford et al., 2021) embeddings. For each masked token, the decoder outputs a distribution over a frozen pretrained VQGAN (Esser et al., 2021) codebook. The model is trained with cross-entropy loss.

At test-time, our model exhibits emerging capabilities such as zero-shot prompting for vision tasks, e.g., performing foreground segmentation with only a textual description of the task without any image examples.

We explore the outcomes of interchanging and combining image and textual prompts in our model. We find that when using both modalities our model achieves improved ICL performance compared to past vision-only approaches (Figure 1), improving average precision by over 10% in Foreground Segmentation, over 4% for single object detection, and closing over 40% of the gap between current ICL approaches to state-of-the-art 1-shot training approaches that utilize supervised base-class training data. Beyond visual recognition, IMProv can be applied to general vision tasks including edge estimation, depth estimation, and conditional image synthesis as shown in Figure 1.

## 2 Prompting Inpainting Models via Images and Text

We start by presenting our IMProv model and how to train it in Section 2.1, and subsequently, in Section 2.2, we explain the approach for prompting the model using visual and textual prompts. Finally, in Section 2.3, we describe the new dataset of images and associated captions we collected.

### 2.1 IMProv - Text Conditional Inpainting Model

We introduce a model with **I**npainting-based **M**ultimodal **Pro**mpting capabilities for **v**ision tasks (IMProv). It receives both text and masked input image as context and outputs a reconstructed image.

Given an input image $x \in \mathbb{R}^{H \times W \times 3}$, a binary mask $m \in \{0, 1\}^{H \times W}$, and a sentence $t \in K \times V$ where $V$ is the vocabulary and $K$ in the sentence length, the goal of our inpainting model $f$ is to generate a new image $y \in \mathbb{R}^{H \times W \times 3}$, with the masked regions filled according to the input image context and the sentence:

$$y = f(x, m, t) \tag{1}$$

Our model $f$ has an encoder-decoder structure like MAE-VQGAN (Bar et al., 2022), where the encoder and decoder are Vision Transformers (Dosovitskiy et al., 2020). In contrast to Bar et al. (2022); He et al. (2021), after every self-attention layer, we add a cross-attention layer between image tokens and textual tokens, thereby effectively allowing each image token to attend to text token:

$$Z_i = \sum_{j=1}^{n} a_{ij} V_j \qquad a_{ij} = \frac{\exp(K_j^T Q_i))}{\sum_{m=1}^{n} \exp(K_m, Q_i))} \tag{2}$$

Where $V$ is the set of textual token values, $K$ is the set of text token keys and $Q$ is the set of image token queries. The resulting output sequence $Z$ represents the attended image features that are most relevant to the text tokens.

**Training.** To train the model, the input image $x$ is split into patches and randomly masked by dropping a fixed percent of the patches (75% in our experiments). Similarly, the input textual sentence is tokenized and every token is mapped to its corresponding CLIP (Radford et al., 2021) embedding. Given the subset of non-masked patches and the textual tokens image, the model is then trained to predict the visual tokens corresponding to masked patches. The model is trained with a cross-entropy loss applied between the model predictions and the corresponding visual tokens for each masked patch. The ground truth visual tokens are obtained by mapping the input image to visual tokens indices using a pre-trained VQGAN encoder (Esser et al., 2021). Formally, our text-conditioned MAE-VQGAN models the distribution $p(z_i|x, m, t)$, where $z_i$ is a visual token from the VQGAN vocabulary that corresponds to the $i$-th image patch.

## 2.2  Multimodal Prompt

At inference time, prompting the trained model can be done via text, via a visual prompt, or by combining both. To prompt the model via visual prompt we apply the same task formulation of Bar et al. (2022) - we form a grid-like image composed of task input-output example(s) (e.g. input images and their segmentation masks), and a new query image, and apply the model to inpaint the corresponding result for the query. To prompt the model via text, we provide to $f$ a description of the task (e.g. "Left: input images, Right: foreground/background segmentation results").

Table 1: **Examples of different textual prompts used for inference with IMProv.**

| Prompt | Full Text Examples |
|---|---|
| No Text | $\phi$ |
| Task | `Image Segmentation` |
| + Location | `Left – input image, right: Black and white foreground background segmentation` |
| + Class Name | `Left – input image, right: Black and white foreground background segmentation of a horse` |

Formally, Let $S = \{(x_i, y_i)\}_{i=1}^n$ be the set of input-output examples where $x_i$ is an image and $y_i$ is the corresponding vision task output, let $t$ be a textual task description and let $x_q$ be a query image. We introduce an arrangement function $g_1$ that arranges $S$ and $x_q$ into a grid (visual prompt), denoted by $x_{vp}$ and provides a mask $m$ for the inpainting function:

$$x_{vp}, m = g_1(S, x_q) \tag{3}$$

Similarly, we have a corresponding arrangement function $g_2$ that generates a textual prompt that is used to instruct the model how to inpaint the image given attributes like the task name, location details, and the image class name:

$$t = g_2(task, loc, class) \tag{4}$$

For example, for the task of image segmentation of an airplane, the output can be "Left: input image of an airplane, Right: corresponding image segmentation". For more examples see Table 1.

The model $f$ is then applied to reconstruct the masked area $m$ given the visual prompt and the task description:

$$y = f(x_{vp}, m, t) \tag{5}$$

When the task description is an empty string, no textual prompt is given and the model has to infer the task solely from the examples of $S$. When $S$ is an empty set, and a textual prompt is given, the model performs zero-shot completion by relying only on the textual instructions.

## 2.3  Image-Text Dataset for Computer Vision

The grid-like visual prompt images inpainted by IM-Prov have a different distribution from natural images. Vision task descriptions (e.g. "left: input, right: segmentation mask"), paired with images, do

Table 2: **Dataset comparison.**

| Dataset | images | papers | with text | source |
|---------|--------|--------|-----------|--------|
| CVF | 78,227 | 20,764 | no | arxiv |
| S2CV | 268,118 | 261,225 | Yes | Semantic Scholar |

not appear often in widely used language-and-vision datasets. Thus, a model that was trained on these datasets will have trouble completing the inpainting task successfully due to the distribution shift. To mitigate this domain gap, we collect a new dataset of figures, paired with their associated captions, extracted from computer vision papers.

Our dataset, The Semantic Scholar Computer Vision dataset (S2CV), is collected from computer vision papers that appear on "Semantic Scholar" website. This website contains papers from 40 conferences and journals from the years 2010 to 2022. We extracted pairs of figures and their captions from each paper on the website, resulting in 1,417,398 pairs. We then filtered out figures that do not include images (e.g. plot of loss curve). Finally, the filtered S2CV dataset includes 268,118 captioned figures and is 3 times larger than the largest existing figures dataset, the Computer Vision Figures dataset (CVF; Bar et al. (2022)). See comparison in Table 2, full details about S2CV in the dataset datasheet (Gebru et al., 2021), and provided examples in Figure 11.

We also extend the existing CVF by repeating its data collection process and extracting the captions of the figures in the dataset. This results in 78,227 image-text pairs. This dataset, CCVF (Captioned-CVF), serves as a baseline in our experiments.

## 3 Experiments and Results

We train a IMProv with a ViT-L backbone on a combination of our CCVF, S2CV dataset and LAION-400M (Schuhmann et al., 2021). During the training process, we create mini-batches by randomly selecting half of the data from the LAION-400M dataset and the other half from CCVF and S2CV, ensuring that the model learns from a diverse set of figure-like images.

We evaluate IMProv on a variety of computer vision tasks. By default, our visual prompt consists of a $2 \times 2$ grid where the bottom-right quarter is masked, the top row contains the input-output example, and the bottom-left image represents the query. The visual example and the textual prompt are defined according to the task (see Section 3.2).

### 3.1 Implementation Details

During training, we utilize images and their associated captions. We follow the resized cropping and flipping augmentations of He et al. (2021) and train on $224 \times 224$ crops. We use AdamW (Loshchilov & Hutter, 2017) optimizer with a learning rate of $2e^{-4}$ and weight decay of 0.05. We train our models on one machine with 8 A100 GPUs with a batch size of 2048 for 150k iterations. Our learning-rate schedule consists of 2k linear warm-up steps followed by a cosine learning rate decay. We use a pre-trained frozen CLIP ViT-L/14 model as our text encoder and a pre-trained VQGAN codebook with a vocabulary size of 1024, provided by Esser et al. (2021) and a spatial dimension reduction of $\times 16$. During training, we drop the text conditioning with a probability of 0.1.

### 3.2 Downstream Computer Vision Tasks

Next, we include the evaluation results of IMProv on a wide range of computer vision tasks. When trained on CCVF/ S2CV and LAION 400M (Schuhmann et al., 2021), IMProv significantly improves ICL performance over a wide range of computer vision downstream tasks when compared to vision-only ICL approaches.

**Foreground Segmentation.** In this task, the goal is to segment the query image into two classes - foreground and background. The input-output example is a random image with its corresponding binary segmentation mask (e.g. black for the background and white for the foreground). We define the textual prompt to be: "Left - input image, right - Black and white foreground-background segmentation of {class}", where the {class} is the class of the foreground object, annotated in Pascal-$5^i$. We follow the evaluation protocol of Bar et al.

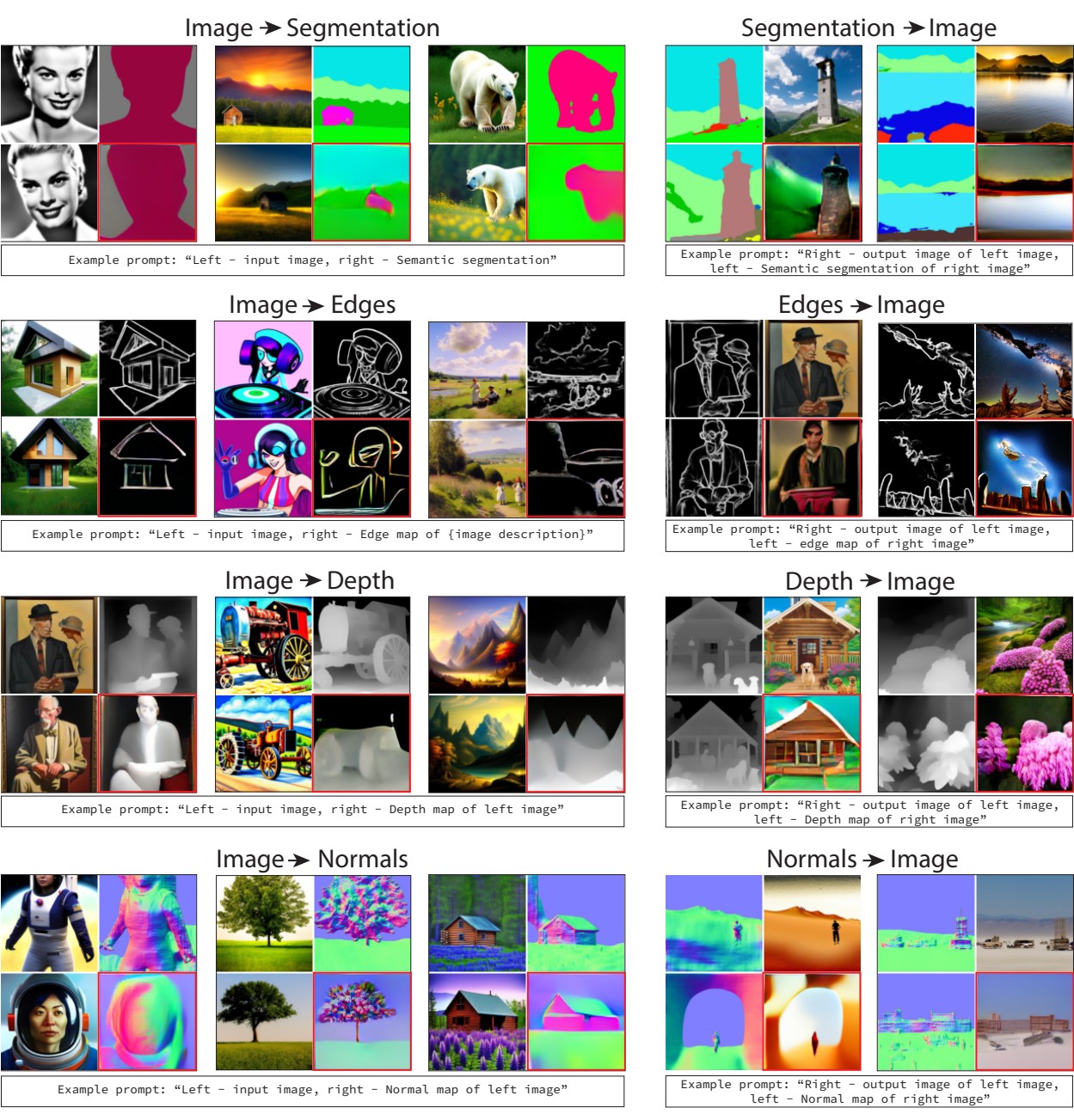

Figure 3: **Multimodal prompting prediction examples.** An example text prompt that was provided to IMProv together with the presented visual prompt appears below them. For each prompt, the result is marked in red. Please see the supplementary material for more results.

(2022) and test IMProv on four splits of Pascal-5$^i$ dataset (Shaban et al., 2017). Results are reported in Table 3.

**Object Detection.** Similar to the task of Foreground Segmentation, our objective here is to perform binary segmentation of the object present in the query image. However, this task is more challenging as the input-output examples contain a rectangle-shaped mask, derived from a bounding box which is less accurate compared to a fine detailed segmentation mask. We define the textual prompt to be: "Left - input image, right - Black and white foreground background segmentation of {class} of rectangle shape" where the {class} is the class of the foreground object. We use the Pascal VOC 2012 dataset (Everingham et al., 2015), which

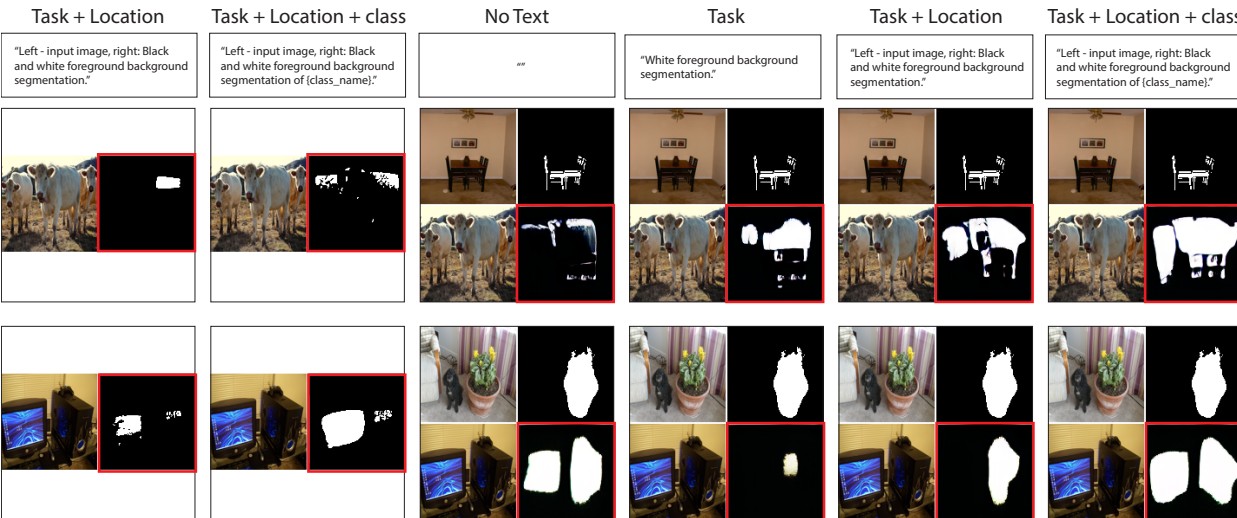

Figure 4: **Detailed textual prompts IMProv performance**. We experiment with textual prompts with varied amounts of detail, e.g., from no text to instructions that include the task, specific locations, and object class names. See examples of full-text prompts in Table 1. Please see the supplementary material for more results.

Table 3: **Comparison between previous visual prompting results to multimodal prompting on computer vision tasks.** For Foreground Segmentation and Single Object Detection, we report the *mIOU* score. For Colorization, we report the *MSE* and *LPIPS*. Training dataset appears in parentheses.

| Model | Foreground Segmentation ↑ | | | | Single Object Detection ↑ | | | | Colorization ↓ | |
|---|---|---|---|---|---|---|---|---|---|---|
| | Split 0 | Split 1 | Split 2 | Split 3 | Split 1 | Split 2 | Split 3 | Split 4 | MSE | LPIPS |
| BEiT (CVF) | 5.38 | 3.94 | 3.20 | 3.29 | 0.17 | 0.02 | 0.14 | 0.16 | 0.60 | 0.70 |
| VQGAN (CVF) | 12.56 | 17.51 | 14.27 | 15.06 | 2.27 | 2.37 | 2.48 | 1.99 | 1.50 | 0.56 |
| MAE (CVF) | 17.42 | 25.70 | 18.64 | 16.53 | 5.49 | 4.98 | 5.24 | 5.84 | **0.43** | 0.55 |
| MAE-VQGAN (CVF) | 27.83 | 30.44 | 26.15 | 24.25 | 24.19 | 25.20 | 25.36 | 25.23 | 0.67 | 0.40 |
| IMProv (S2CV + LAION) | **42.58** | **44.81** | **40.73** | **33.72** | **30.03** | **30.73** | **29.8** | **31.32** | 0.57 | **0.34** |

consists of images along with their associated detection boxes. Our results are reported in Table 3 in terms of the mean Intersection over Union (mIOU) metric.

**Colorization.** The goal is to map a gray-scale image to a color image. The example pair is a gray-scaled image and the corresponding color image. We define the textual prompt to be: "Colorization results: Left - input image, Right - Colorized image of class" where the {class} is the class of object present in the image. We randomly sampled 1000 example pairs and image query from ImageNet (Russakovsky et al., 2015) validation set and converted them to gray-scale to obtain gray-scale and color version for each image. MSE and LPIPS (Zhang et al., 2018) Results are reported in Table 3.

**Other Tasks.** We evaluate our models on the dataset created by Wang et al. (2023c), which includes around 310k image-caption pairs that were automatically annotated by using state-of-the-art pre-trained models for a wide range of vision tasks. Specifically, each image is annotated with depth and normal maps obtained from Midas (Ranftl et al., 2022), segmentation maps obtained from Uniformer (Li et al., 2022), and object boundary maps detected by HED (Xie & Tu, 2015). For each vision task X, we evaluate our model on two tasks - X-to-images and images-to-X. As each task has a different evaluation metric, and as our model produces image outputs, we simplify the evaluation by comparing the generated image to the rendered annotation of the task by calculating LPIPS (Zhang et al., 2018). We report the results in Table 5 and plot qualitative results in Figure 3.

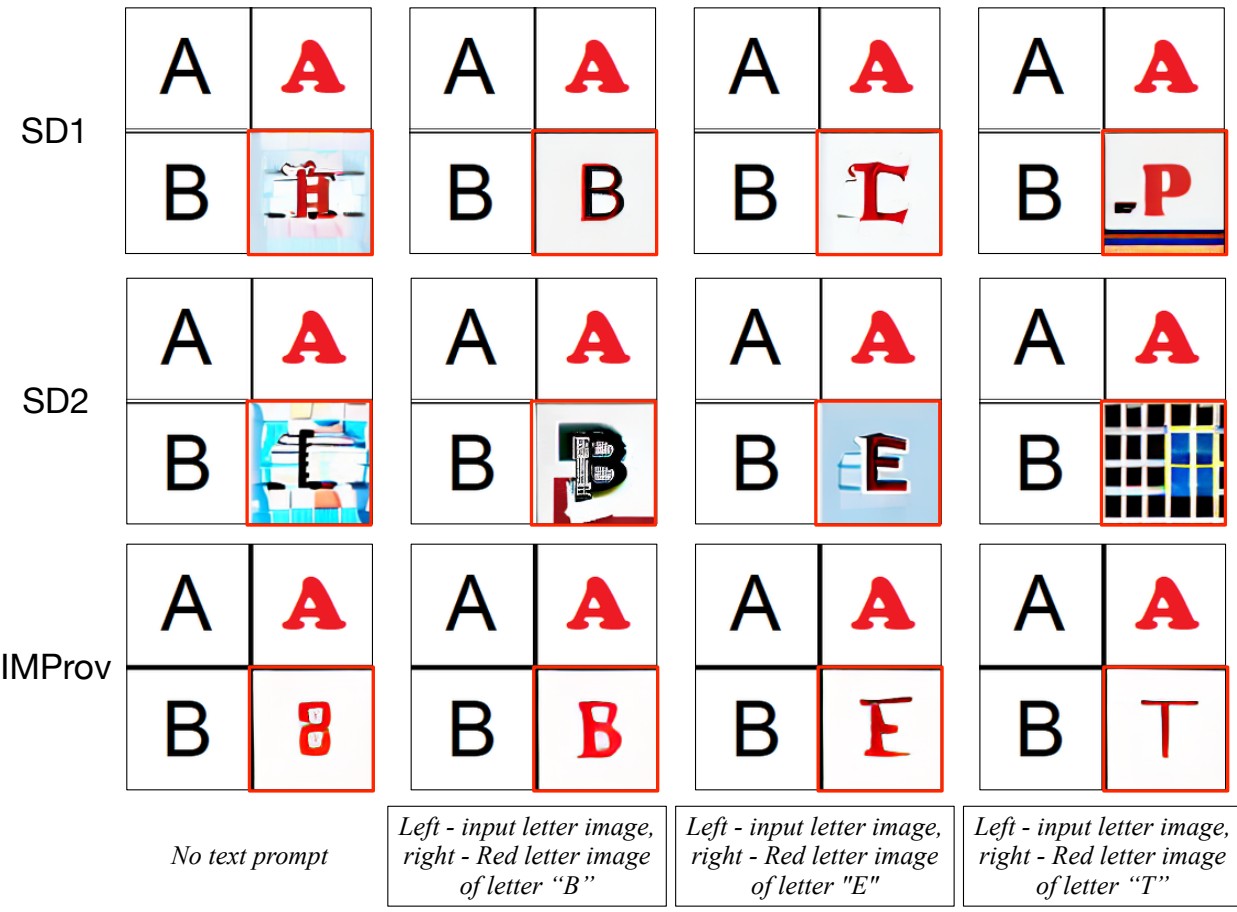

Figure 5: **Textual prompts effect on text inpainting.** When there is an inconsistency between the textual and visual prompt, the model may follow the textual prompt. IMProvcan better follow both visual and text instructions compared to Stable Diffusion (SD) on letter generation.

## 4 Analysis

**Dataset Ablation.** We report our results and compare them with prior works in Table 3. IMProv trained on a combination of LAION-400M and our S2CV dataset outperforms the prior work (Bar et al., 2022) trained solely on the CVF dataset by more than ∼ 12 points in mIOU. It demonstrates IMProv could benefit from training on additional amounts of unlabeled images.

**Textual Prompts Ablation.** We experiment with textual and visual prompts that have different relevance to the query image and task. For the visual prompt, we choose the input-output examples using three different retrieval strategies: (1) *Random*

Table 4: **Text helps.** Adding textual prompts to "*Random Class*" visual prompts improves Foreground Segmentation.

| Model | Avg. |
|---|---|
| MAE-VQGAN (CVF) | 23.52 |
| IMProv(CCVF) | 26.13 |
| IMProv(CCVF + LAION) | 36.29 |

*Class*, a random example pair in which the class of the foreground object is chosen randomly, (2) *Same Class*, where a random example pair of the same class is chosen randomly, and (3) The example is chosen via *Nearest Neighbor* from all the images with the same foreground object class, using the model from Zhang et al. (2023). We evaluate our IMProv model on Pascal $5^i$ with and without a textual prompt that contains Task, Location, and Class Name.

Firstly, we compare against Bar et al. (2022) under (1) *Random Class* visual prompt setting and report results in Table 4. In this setting, the visual prompts describe the task (e.g., segmentation) but are not

Table 5: **Training on additional annotated data improves ICL.** We train IMProv on unstructured and structured data. We find that adding structured and fully annotated data during training can help ICL performance on 8 *held-out* computer vision tasks.

| | Depth→ Image | Image→ Depth | HED→ Image | Image→ HED | Seg→ Image | Image→ Seg | Normals→ Image | Image→ Normals |
|---|---|---|---|---|---|---|---|---|
| Supervised ICL (InstructPix2Pix) | 0.65 | 0.60 | 0.59 | 0.62 | 0.64 | 0.61 | 0.62 | 0.55 |
| IMProv (S2CV+LAION) | 0.61 | 0.52 | 0.51 | 0.46 | 0.59 | 0.50 | 0.56 | 0.48 |
| IMProv (S2CV+LAION+InstructPix2Pix) | **0.55** | **0.43** | **0.47** | **0.37** | **0.54** | **0.46** | **0.51** | **0.44** |

curated from the same class (the setting in Table 3), or chosen via nearest neighbors as in Zhang et al. (2023). Using non-curated visual prompts is most realistic, as finding a perfectly aligned visual example might be as hard as solving the original input. The result shows that conditioning on text improves average mIoU by 3 points when using reasonable non-curated visual prompts. Moreover, IMProv trained on a combination of LAION and our CCVF dataset further boost the mIoU by 10 points.

In Figure 6 we plot the results under different textual prompts. We find that the textual prompts play a big role in the performance of the model (see Figure 6). To dive deeper into the effect of textual prompt, we plot the relation between textual prompt and visual prompt in Figure 4. It shows adding text prompts improves the results for any type of visual prompt, from the least related Random Class examples to the most relevant Nearest Neighbors examples. In addition, we find that by using text, it is possible to achieve similar performance with lower-quality visual examples (using the Same Class example rather than the Nearest Neighbor (Zhang et al., 2023)). Similarly, higher-quality visual examples improve the results for all the tested textual prompts. Interestingly, the results suggest a trade-off between the two modalities - high-quality textual prompts can alleviate the need for carefully chosen visual prompts, and vice versa.

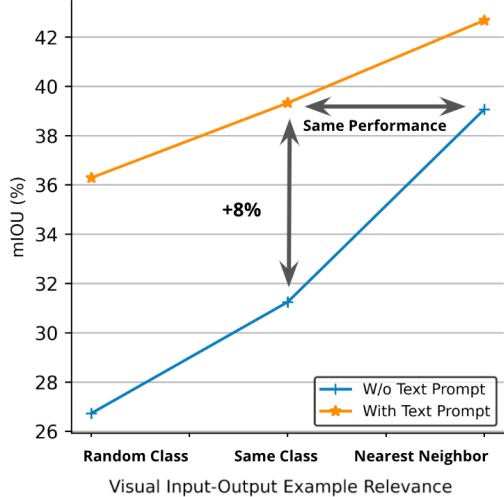

Figure 6: Using textual prompts improves performance and reduces the need for careful selection of visual prompts.

Moreover, as shown in Figure 5, when the textual prompt is inconsistent with the visual example, the model may follow the more certain textual instruction. We include additional results for different combinations of visual and textual prompts in the Supplementary Material.

**Does Structured Data Improve In-Context Learning?**
The key insight of our approach is to train IMProv on *unstructured* data in a fully unsupervised manner without parsing the image or the text. Here we experiment with training IMProv on additional *structured* data in a fully supervised manner. We use the dataset of Brooks et al. (2022), which consists of 310k input-output image editing pairs and their corresponding descriptions. For training, we use random pairs as our input-output examples. We embed them into a grid structure, in a similar manner to the structure we use at test-time. The grid images that we construct for training consist of $1 \times 2$ and $2 \times 2$ images by randomly selecting 1 or 2 input-output examples for each caption in the original dataset.

We test our models on a held-out set of vision tasks. As shown in Table 5, we find that training on structured supervised data alone leads to poor generalization and ICL performance on the test tasks. Training on both unstructured S2CV and LAION-400M, together with the structured data improves ICL results on the test tasks compared to our base model.

**Comparison to Finetuning and Few-Shot Baselines.** We compare IMProv to classic 1-shot baselines, which we view as an upper bound of our approach. Approaches like FWB (Nguyen & Todorovic, 2019) and CyCTR (Zhang et al., 2021) utilize a fully labeled base classes train set (2086 to 5883 on different Pascal $5^i$ splits) with architectures that are optimized for foreground segmentation (e.g, by utilizing higher resolutions). We also compare to MAE-VQGAN (Bar et al., 2022) that performs visual prompting without text and to

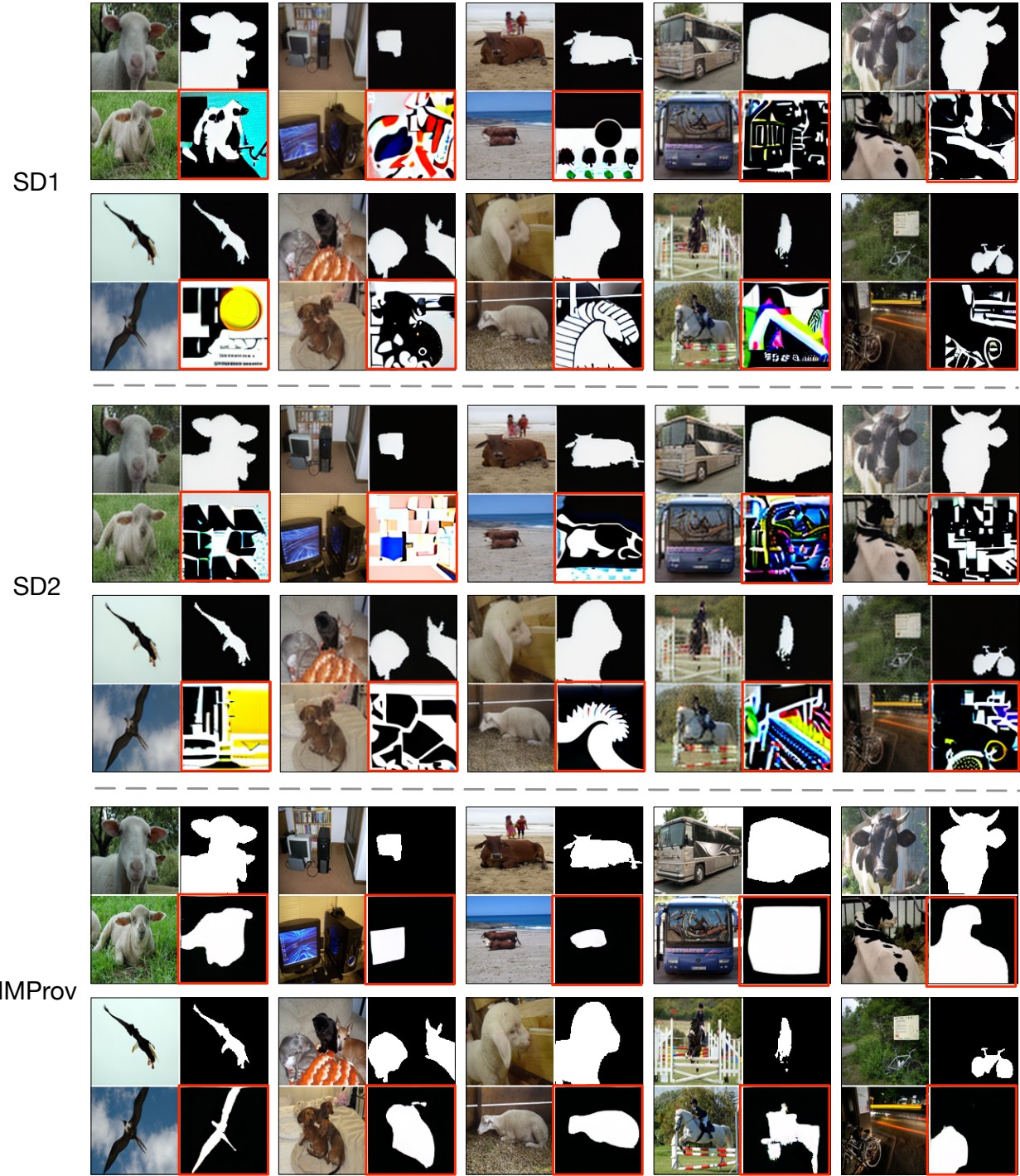

Figure 7: **Compare to Stable Diffusion (SD) on Segmentation.** The result is marked in red.

finetuning baselines with $K = \{1, 4, 16\}$ training examples for each target class. The results in Table 6 indicate that IMProv closes over 40% of the accuracy gap between MAE-VQGAN to supervised one-shot approaches. This demonstrates the potential of our approach to scale with more data.

**Comparison to existing text-to-image works.** We also compare our IMProv against state-of-the-art text-image generative models, i.e. Stable Diffusion Rombach et al. (2021). We input the same text prompt and visual prompt to Stable Diffusion version 1 (SD1) and 2 (SD2) inpainting models, with 50 inference steps and 7.5 classifier free guidance scale. The quantitative results are shown in Figure 7. Compared to IMProv, SD fails to generate black/white segmentation mask, and couldn't leverage visual prompt to

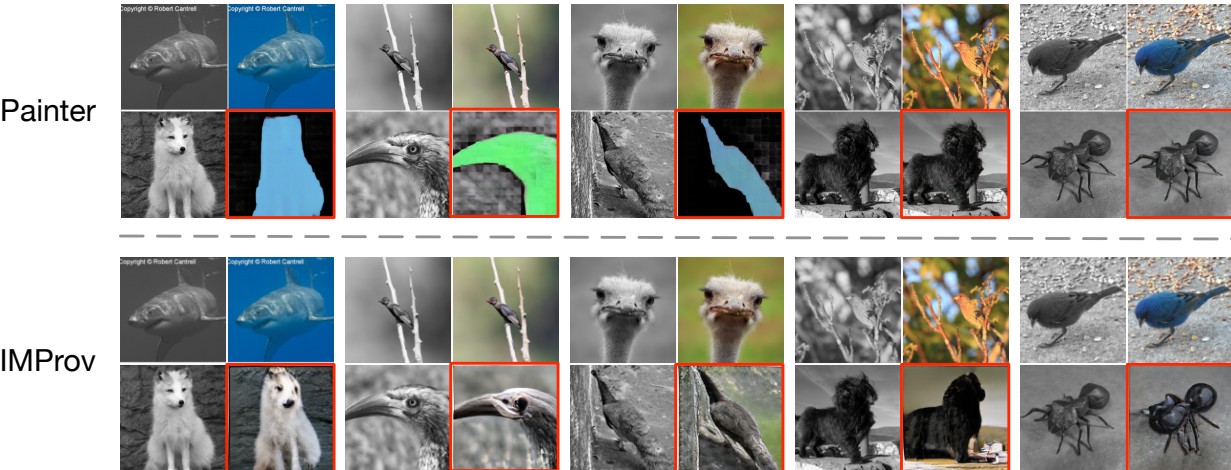

Figure 8: **Comparison with other visual prompt methods**. We compare IMProv with PainterWang et al. (2023a) in the task of colorization. Both models are not trained explicitly with colorization supervision. Wang et al. (2023a) fails in some images, generating segmentation masks instead, while ours could generate reasonable colorized images.

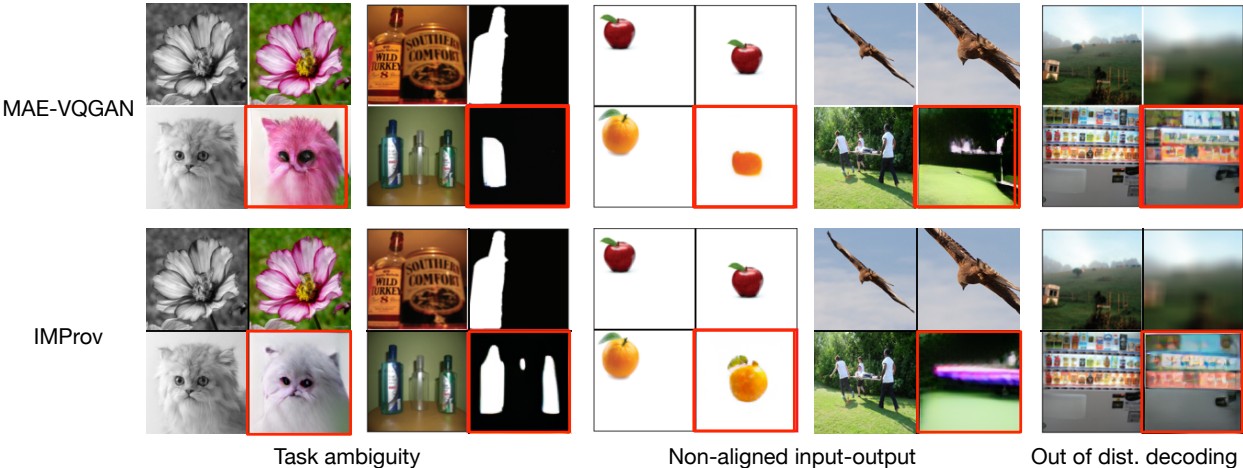

Task ambiguity        Non-aligned input-output        Out of dist. decoding

Figure 9: **Failure case analysis**. Compare to Bar et al. (2022), IMProv could succeed on some cases to some extend e.g. "Task ambiguity". But we still fail on one of the "Non-aligned input-output" cases.

find the corresponding objects. We also compare them on the letter generation task in Figure 5. Although SD sometimes could generate the correct letter, it still fails to follow the visual prompt to generate white background and red font color. Moreover, as shown in Figure 5, when the textual prompt is inconsistent with the visual example, the model may follow the more certain textual instruction.

**Compare to supervised approach**. Compared to a supervised prompting approach like Painter(Wang et al., 2023a), IMProv can generalize to a larger variety of tasks. We demonstrate this in Figure 8, showing that while Painter fails to adapt to the colorization task, IMProv performs reasonably well. We additionally report the comparison of Painter and our method in Table 7. Our model outperforms Painter by a large margin. It's also worth noting that changing the training dataset from CCVF to S2CV improves the mIoU significantly, which further indicates the effectiveness of our proposed S2CV dataset.

Table 7: Segmentation results on the FSS-1000 dataset.

| Model | FSS-1000 mIoU |
|---|---|
| VisualPrompt | 58.3 |
| Painter | 62.3 |
| IMProv (CCVF) | 62.8 |
| IMProv (S2CV) | 68.9 |

Table 6: **Comparison to fine tuning and classic 1-shot segmentation baselines.** MAE-VQGAN image query and output resolution is 111×111. CyCTR and FWB resolution is 473×473 and 512×512, both approach utilize Pascal $5^i$ labeled baseclasses data.

| Pretraining | # Labeled Images | # Shots | Model | Split 0 | Split 1 | Split 2 | Split 3 |
|---|---|---|---|---|---|---|---|
| | 1 | 1 | | 11.1 | 13.4 | 13.0 | 12.3 |
| Unlabeled ImageNet | 4 | 4 | Finetuned MAE (He et al., 2021) | 12.9 | 15.8 | 14.3 | 15.0 |
| | 16 | 16 | | 13.7 | 16.1 | 16.8 | 17.1 |
| CVF + IN | 1 | 1 | MAE-VQGAN (Bar et al., 2022) | 32.5 | 33.8 | 32.7 | 27.2 |
| CCVF + LAION | 1 | 1 | IMProv | 45.6 | 46.6 | 45.3 | 39.1 |
| S2CV + LAION | 1 | 1 | IMProv | **49.1** | **49.7** | **45.5** | **42.1** |
| **Labeled** Pascal 5i (Segmentation masks) | 2086 − 5883 | 1 | FWB (Nguyen & Todorovic, 2019) | 51.3 | 64.5 | 56.7 | 52.2 |
| | | 1 | CyCTR (Zhang et al., 2021) | 67.2 | 71.1 | 57.6 | 59.0 |

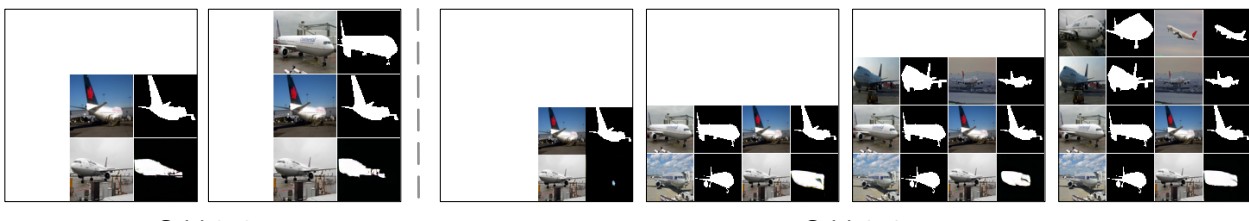

Grid 3x3                    Grid 4x4

Figure 10: **Different grid size**. IMProv also support different grid size other than 2x2 grid. We show segmentation results with 3x3 grid and 4x4 grid with different number of visual prompts. Note, the white region are discarded and not input to the model.

**Different grid size** With a 2x2 grid, only a visual prompt is applicable. We also report different grid sizes with different numbers of visual prompts. We report the results on Pascal VOC segmentation in Table 8 and Figure 10. In each row, keeping grid size fixed, as we increase the number of visual prompt

Table 8: **Different grid size.**

| number of visual prompt | 1 | 2 | 3 | 5 | 7 |
|---|---|---|---|---|---|
| Grid 2x2 | 42.68 | - | - | - | - |
| Grid 3x3 | 37.21 | 39.78 | - | - | - |
| Grid 4x4 | 16.42 | 18.34 | 30.78 | 32.56 | 33.11 |

examples, the mIoU increases. On the other hand, due to the limitation of 224x224 resolution, when the grid size is larger, image in each shot becomes smaller, so one-shot accuracy drops. We acknowledge that the current model could only take a fixed image size, as many other pre-train image encoders. However, if more computation budgets are available, this issue could be mitigated with higher training resolution.

**Failure case analysis** We compare with failure cases of Bar et al. (2022) in Figure 9. For some of the cases, our IMProv could successfully address them with text prompts, e.g. the cat colorization and bottle segmentation of "Task ambiguity". And our model could successfully generate the image that moves the orange to the center, though fails the other Non-aligned input-output example.

# 5    Related Work

**Prompting in NLP.** The ability to prompt a language model to solve a specific task, also known as ICL, is a recently discovered property of generative language models that were trained on a large corpus of text (Brown et al., 2020; Touvron et al., 2023; Chowdhery et al., 2022; Bubeck et al., 2023). Brown et al. (2020) have shown that existing NLP tasks can be described in unstructured text, and then fed into a large language model to complete the missing part without any finetuning (Radford et al., 2019; Brown et al., 2020). More recently different approaches to prompting have emerged including Prompt Engineering (Brown et al., 2020; Lu et al., 2021), Prompt Ensembling (Jiang et al., 2020), Prompt Prefix Tuning (Li & Liang, 2021; Lester et al., 2021), and Chain of Thought Prompting (Wei et al., 2022). The Flamingo (Alayrac et al., 2022) model extends language-only models, and conditions the model on both image and text. Our approach is different in that our model outputs pixels and not text. Therefore, it is suited to solve a variety of computer vision tasks that can be represented in pixel-space, like semantic segmentation or image colorization.

**Visual Prompting**. Recently, multiple papers have proposed methods to visually prompt computer vision models (Bahng et al., 2022; Jia et al., 2022; Bar et al., 2022). Bahng et al. (2022) proposes to add noise tensor to the input image to adapt the model to different tasks, while Jia et al. (2022) proposes to append learned tokens to Vision Transformers (Dosovitskiy et al., 2020), which draws motivation from prefix tuning in NLP (Li & Liang, 2021). These two approaches are trained on supervised data and thus struggle to scale and generalize to new tasks. Bar et al. (2022) takes a different approach and trains on unstructured crops from computer vision paper figures. According to this approach, visual prompting is viewed as an Image Inpainting task by creating an image grid containing input-output examples and new image input. The goal of the inpainting model is then to complete the output in a way that is consistent with the input. We follow a similar definition of visual prompt as in (Bar et al., 2022), however, we propose to condition the model on textual input as well which might help to solve ambiguities in the task description and can more efficiently guide the visual model toward performing the desired task.

**Image Inpainting and Image Synthesis**. Early image inpainting methods relied on the input image itself for inpainting (Efros & Leung, 1999; Bertalmio et al., 2000; Criminisi et al., 2004; Barnes et al., 2009), whereas more recent works leveraged image datasets for this purpose (Hays & Efros, 2007; Pathak et al., 2016; Yang et al., 2017; Liu et al., 2018b;a). Lately, diffusion models have demonstrated large success in image inpainting and image synthesis (Ramesh et al., 2022; Rombach et al., 2021), as well as other popular transformer-based methods (Chen et al., 2020; Yu et al., 2021b; Esser et al., 2021; Yu et al., 2021a; Chang et al., 2022). Few of these approaches rely on discrete latent codebooks which induce a distribution over possible completions (Van Den Oord et al., 2017; Ramesh et al., 2021; Esser et al., 2021; Yu et al., 2021a; Chang et al., 2022). For instance, Esser et al. (2021); Yu et al. (2021a) proposed to synthesize images using an autoregressive model on a codebook representation, while Chang et al. (2022) applied iterative parallel decoding of the tokens. Few approaches also support image synthesis with text conditioning - MUSE (Chang et al., 2023), for example, is a transformer-based model that applies cross-attention from image embeddings (VQGAN (Esser et al., 2021)) to the text embeddings extracted from a pre-trained model (e.g. T5 (Raffel et al., 2020)) to condition on text. Our model is conceptually similar to MUSE (Chang et al., 2023), however, we focus on inpainting grid-like visual prompts that require reasoning across multiple sub-images and input text.

**Few-Shot Learning.** In this setting, the algorithm is trained on a labeled dataset of base classes, from which it should transfer to a set of novel classes given only a few training examples (like 10 or 30) (Nguyen & Todorovic, 2019; Kang et al., 2019; Liu et al., 2020; Wang et al., 2020; Yang et al., 2020; Tian et al., 2020; Zhang et al., 2021; Bar et al., 2021). Unlike Few-Shot approaches, here we do not assume access to a large training set of base-classes, and our architecture is not task-specific. Our approach is Few-Shot only in the sense that the visual part of our prompt usually contains one or two task examples.

## 6  Discussion

We presented an approach for multimodal prompting of inpainting models. To unlock in-context learning capabilities in such models, we had to collect a specific dataset of figures with associated captions. To further scale this approach, we believe that other sources of unstructured data - visual, textual, and multimodal - should be incorporated during the training phase. To understand the feasibility of such a data collection effort, one must be able to predict and quantify the effect of different dataset sizes and types on the downstream in-context learning capabilities of models trained on them. We plan to investigate it in future work.

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

## Appendix

We include more information about the experimental study, as well as the Captioned Computer Vision Figures (CCVF) dataset datasheet.

## A    Broader Impact Statement

Using In-Context Learning to solve various vision tasks using one model has the potential to reduce the cost of training and inference of future models, facilitate access to this technology, and democratize its exploration. Nevertheless, while scaling the training dataset to a large corpus of non-curated text-image pairs improves the in-context learning results, it might also introduce potential biases that must be mitigated prior to any commercial deployment of the model. A comprehensive discussion about ethical considerations can be found in Schuhmann et al. (2022).

## B    Experiments and Results

We train IMProv on CCVF/S2CV and LAION 400M and evaluate in-context learning. We first visualize the S2CV as shown in Figure 11. We then explain the details of our experiments as follows.

**Multimodal Prompting.** Inspired by Zhang et al. (2023), we experiment with prompts that have different relevance to the input query image, we include visualization examples in Figure 12.

For the visual prompt, we use five different retrieval strategies for choosing input-output example pairs from Pascal-$5^i$:

- No input-output visual example ("*No Example*")

- A random input-output pair in which the class of the foreground object is different from the class of the foreground object ("*Different Class Random Sample*")

- A random input-output pair with the same foreground object class as in the query image ("*Same Class Random Sample*")

- The nearest neighbor input-output pair in CLIP embedding space Radford et al. (2021) retrieved from all the images with the same foreground object class ("*Same Class CLIP NN*")

- The nearest neighbor retrieved from all the images with the same foreground object class according to the model provided by Zhang et al. (2023) ("*Same Class Zhang et al. (2023) NN*"). This strategy was trained to optimize in-context learning results.

For the text prompt, we experimented with three prompting variations:

- No text prompt

- Text prompt with location and task information but without class - "Left - input image, right - Black and white foreground/background segmentation"

- Text prompt with location, task and class information - "Left - input image, right - Black and white foreground/background segmentation of {class}"

Table 10 and Figure 4 present quantitative and qualitative results for different combinations of visual and textual prompts. We find that more informative textual prompts improve the results for all visual prompts. Similarly, higher-quality visual examples improve the results for all the tested textual prompts. Interestingly, the results suggest a trade-off between the two modalities - high-quality textual prompts can alleviate the need for carefully chosen visual prompts, and vice versa. We argue that using non-curated visual prompts is most realistic, as finding a perfectly aligned visual example might be as hard as solving the original input.

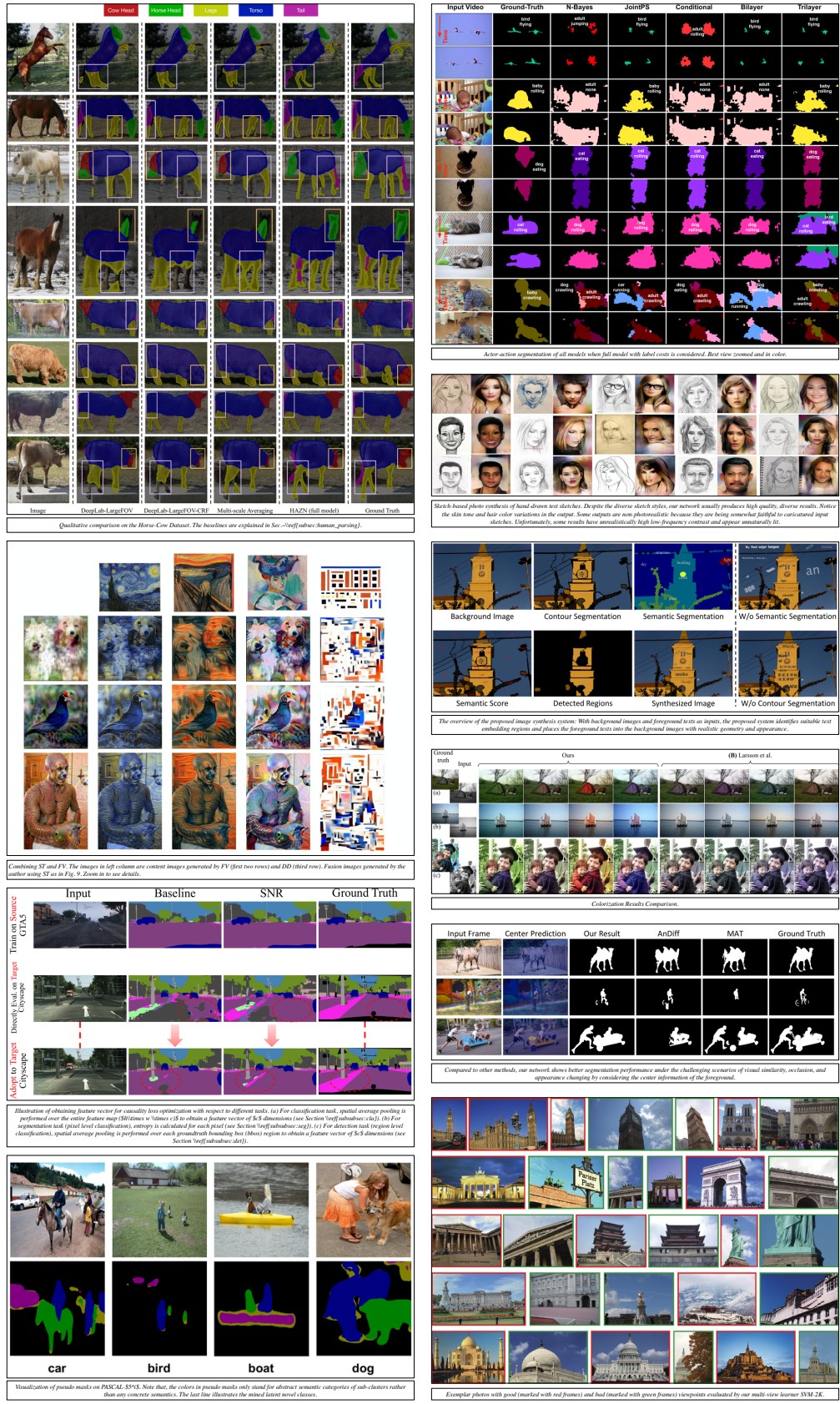

Figure 11: **Image-Text pair examples in S2CV**.

So in Table 9 we report the mIoU on Pascal VOC segmentation, where we compare against Bar et al. (2022) under "*Different Class Random Sample*" visual prompt setting. In this setting, the visual prompts describe the task (e.g., segmentation) but are not curated from the same class (as in Table 2), or chosen via nearest neighbors. The result shows that conditioning on text significantly improves the prompting performance when using reasonable non-curated visual prompts.

Table 9: **Improvement of textual prompt.** Comparison under "*Different Class Random Sample*" visual prompting setting, IMProv outperforms MAE-VQGAN with textual prompt.

| Model | Split 0 | Split 1 | Split 2 | Split 3 | avg |
|---|---|---|---|---|---|
| MAE-VQGAN (CVF) | 24.66 | 25.15 | 24.36 | 19.91 | 23.52 |
| IMProv(CCVF) | 26.15 | 27.38 | 29.37 | 21.62 | 26.13 |
| IMProv(CCVF + LAION) | 37.09 | 40.68 | 36.91 | 30.49 | 36.29 |

Table 10: **Visual and Textual Prompts Combination.** We evaluate our model on different combinations of visual prompts and textual prompts, with varying relations to the query images.

| | | | | Visual Prompt Type | | | |
|---|---|---|---|---|---|---|---|
| Visual Prompt | Text Prompt | w/ class name | No Example | Different Class Random Sample | Same Class Random Sample | Same Class CLIP NN | Same Class Zhang et al. (2023) NN |
| | ✓ | | 18.39 | - | - | - | - |
| | ✓ | ✓ | 18.75 | - | - | - | - |
| ✓ | | | - | 26.73 | 31.25 | 38.14 | 39.07 |
| ✓ | ✓ | | - | 34.50 | 36.62 | 41.30 | 42.17 |
| ✓ | ✓ | ✓ | - | 36.29 | 39.33 | 41.99 | 42.68 |

**Trade-off between the number of visual prompt and textual prompts** We plot the mIoU w.r.t the number of support in Figure 13. We run the experiments under grid 4x4 setting in Table 8, with "Random Class" support image. Similar to Bar et al. (2022), as we increase the number of support visual examples, mIoU goes up. It is worth noting that when there are only 1 or 2 examples, the text prompt does not help. It is because the model couldn't generate meaningful segmentation with a small number of visual supports due to the resolution (see Figure 10 for details). Under the setting that visual support number greater than 3, the text prompt consistently improves the results. It implies that our text prompt is orthogonal to the number of visual support pairs.

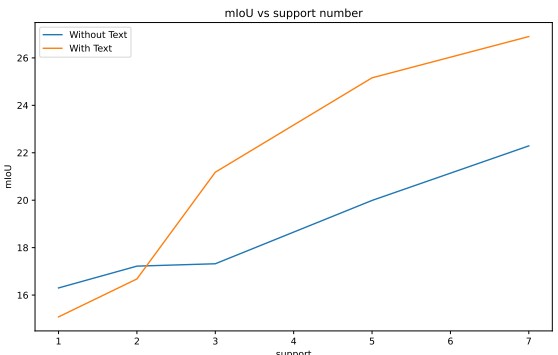

Figure 13: **Trade-off between the number of visual prompt and textual prompts**.

**Other Vision Tasks**. We provide more qualitative results for both Image-to-X task and X-to-Image task in Figure 14 and Figure 15 respectively, "X" can be any of the following: segmentation, edges, depth and normal.

**Generalizability to new vision tasks** Our model was never trained on specific vision tasks, and instead was trained on unstructured and non-annotated figures from computer vision papers. The data we train on is not constructed explicitly for specific tasks, and even if it is trained on some task-specific figures, it is usually not presented the way we test our model, as we randomly crop the figures during training. Therefore, we believe the test task is generalized from different task combination instead of replicating from training task. Similarly to(Bar et al., 2022), our model can generalize to unseen diverse tasks as shown in Figure 16.

**Dataset Ablation** We additionally report the results of IMProv(S2CV+LAION) in the Table 11 under the same "Random Class" setting. We report the average mIoU over 4 splits. IMProv(S2CV + LAION) outperforms IMProv(CCVF + LAION) by 1.7 points, which justifies the effectiveness of our proposed S2CV dataset.

Table 11: **Dataset ablation.**

| Model | Avg. |
|---|---|
| IMProv(CCVF + LAION) | 36.29 |
| IMProv(S2CV + LAION) | 38.07 |

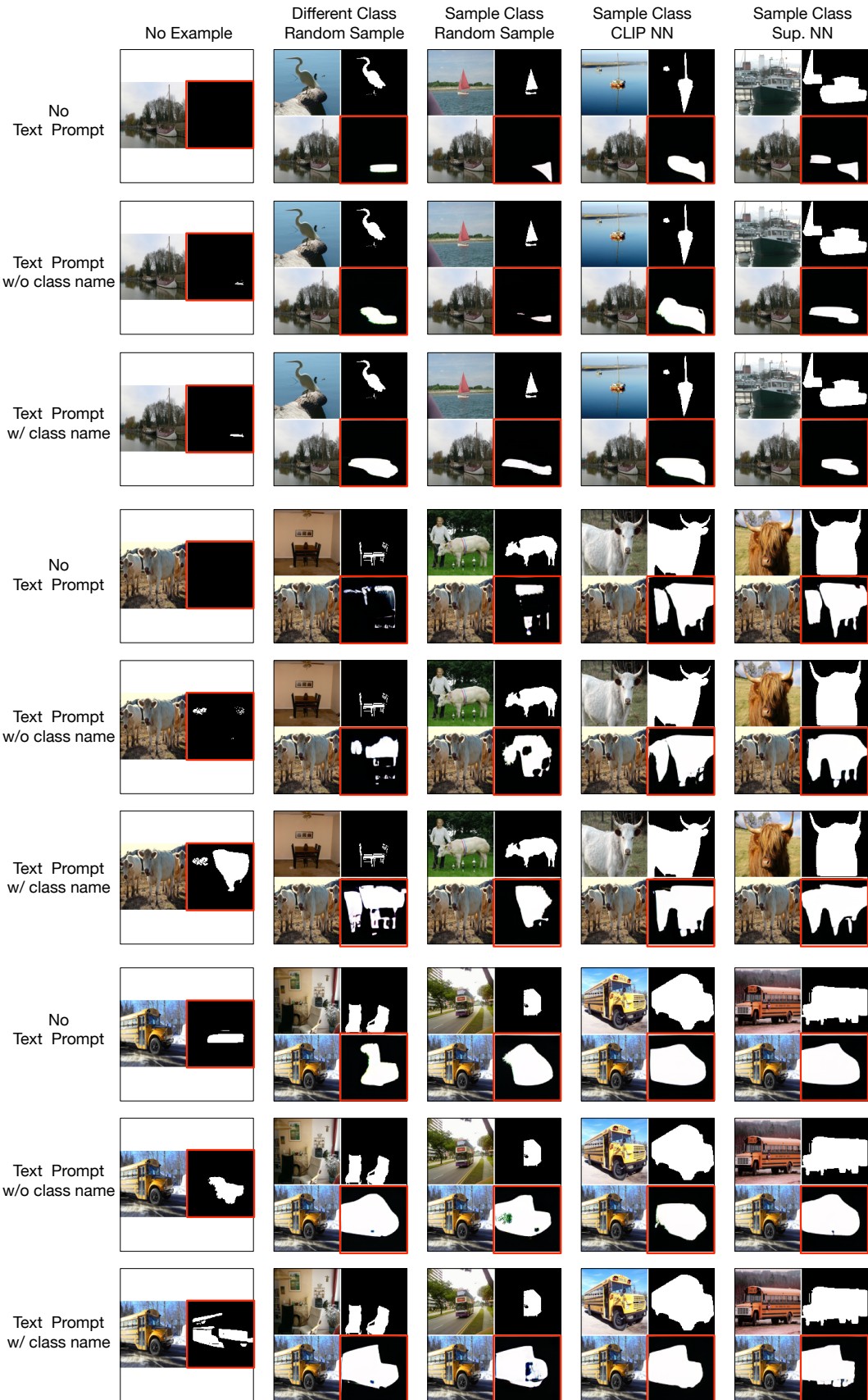

Figure 12: **Visual and Textual Prompts Combination.** Each column represents visual prompt of different relevance. The result is marked in red.

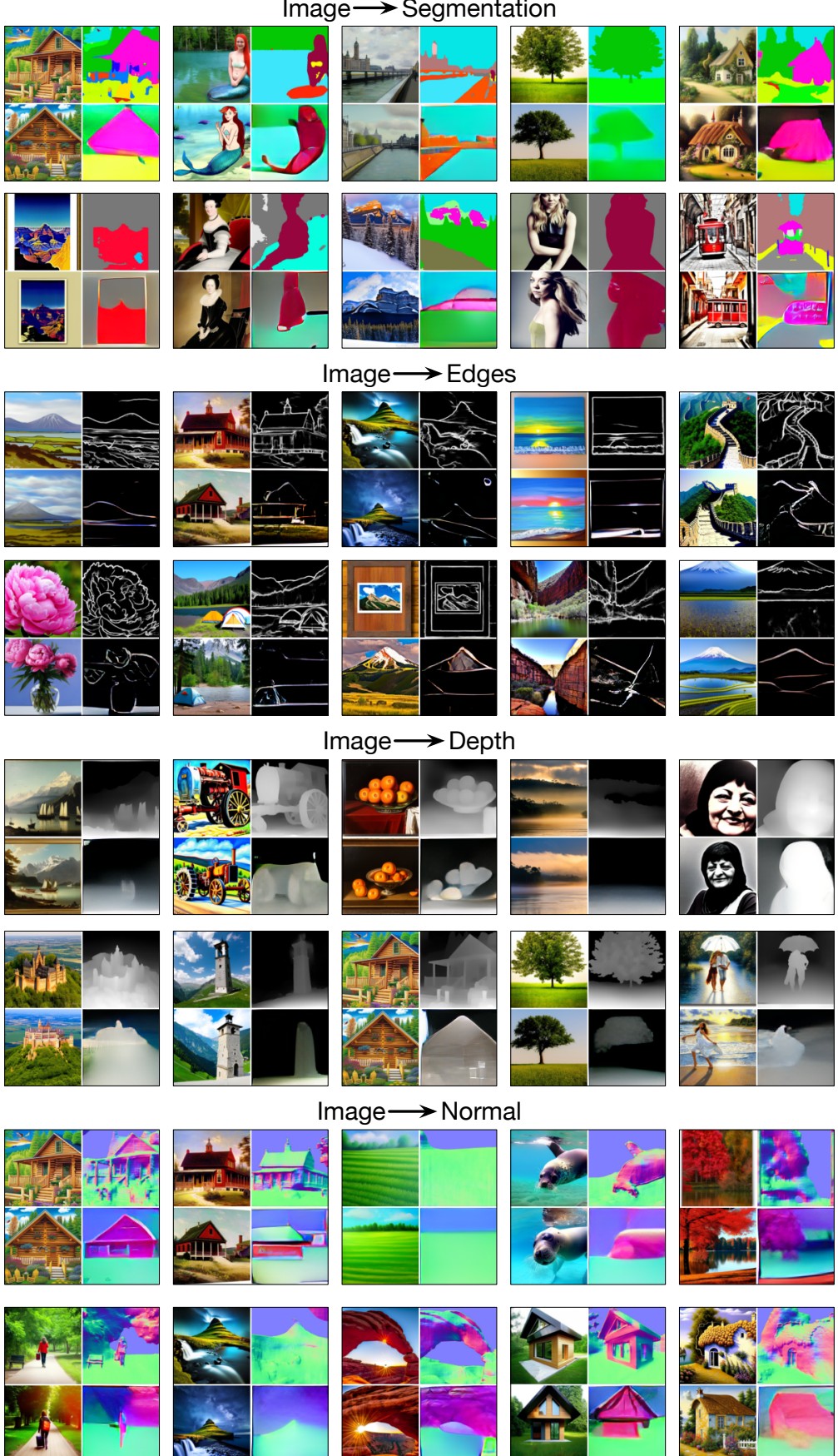

Figure 14: **Image-to-X results.** The result is marked in red.

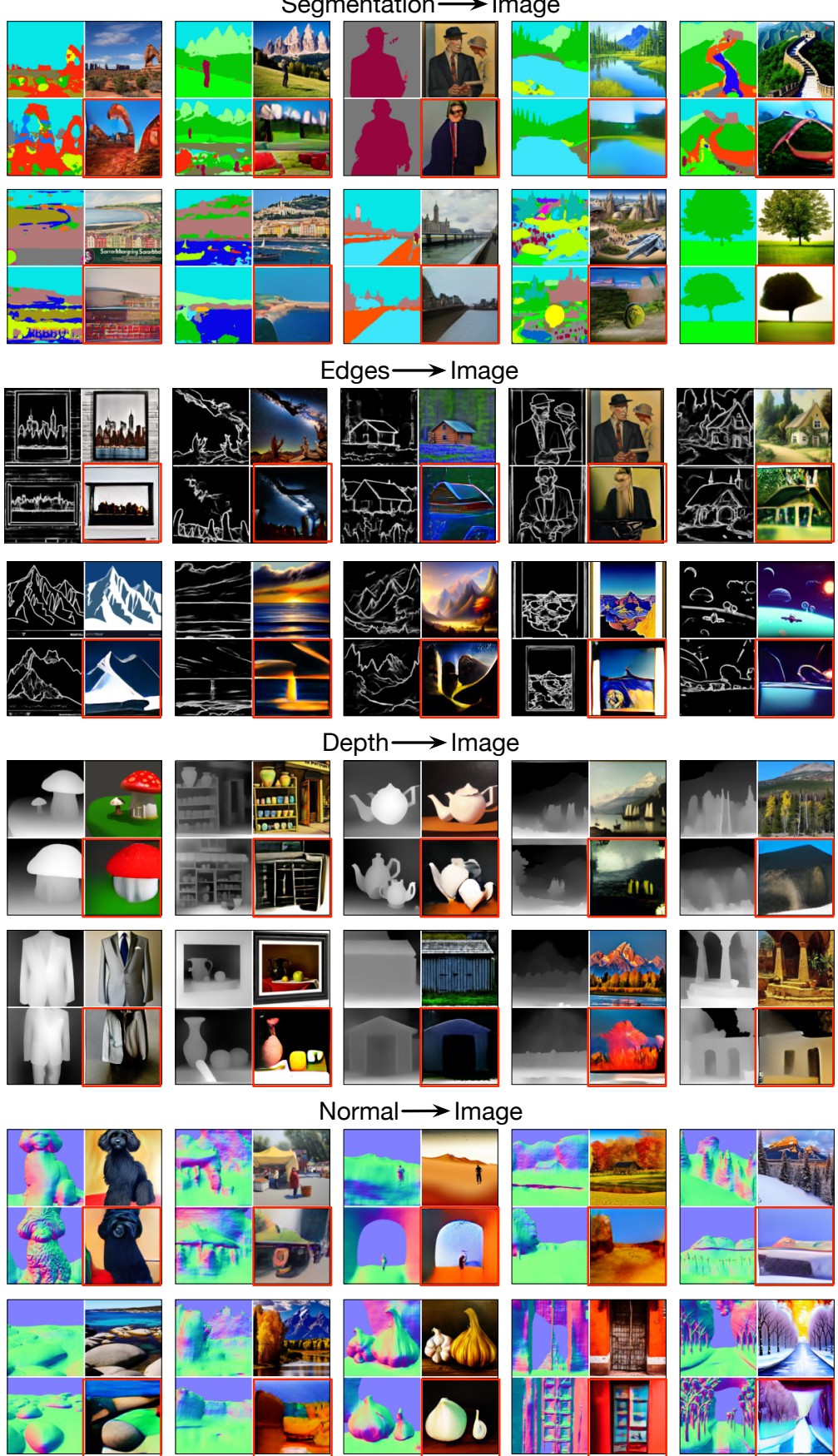

Figure 15: **X-to-Image results.** The result is marked in red.

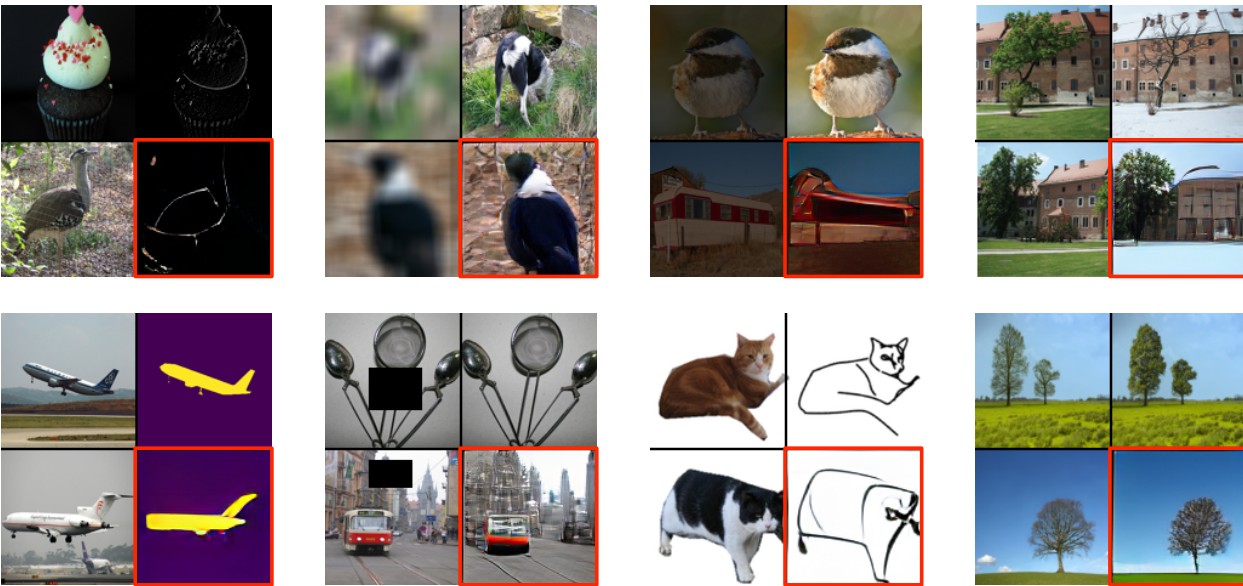

Figure 16: **More vision tasks**. We show IMProv is capable of other vision tasks, including edge detection, debluring, image enhancing, style transfer, image-to-sketch.

