# OpenReview forum: "IMProv: Inpainting-based Multimodal Prompting for Computer Vision Tasks"
_TMLR — Accepted by TMLR_

### Review · Reviewer_UUw9 · 2024-07-16

**Summary Of Contributions:**

This paper proposes a generative model that is able to in-context learn visual tasks from multimodal prompts. The inputs are a textual description of a visual task, a few input-output visual examples, or both, the model in-context learns to solve it for a new test input.

Experiments show that training this proposed model with text conditioning and scaling the dataset size improves in-context learning for computer vision tasks by a large margin. Extensive experiments demonstrate that vision and language prompts are complementary.

**Audience:**

Yes

**Broader Impact Concerns:**

The reviewer did not see any obvious ethical concerns.

**Claims And Evidence:**

Yes

**Requested Changes:**

In general, authors provide a simple yet effective approach to address how to apply in-context learning to vision tasks.

The collected dataset can contribute to the multimodal understanding community if it can be open sourced.

Based on the above weakness, the concerns are about the unfair comparisons, limitations of this approach, and the main contribution.

**Strengths And Weaknesses:**

**Strengths**

1. This paper is clearly written and easy to understand.
2. There are extensive ablation studies to compare different training factors.

**Weakness**

1. The main concern is that the comparisons are not fair. In Table 3, all the baselines are trained with CVF, while the proposed model is trained with a much larger training set (i.e., S2CV+LAION). The same is true for Table 6.

2.  While authors show that providing examples is beneficial to the model inference, the vision encoder takes a fixed image size. This may lead to a degraded performance when increasing the number of visual prompts. Such issue may limit the practical usage of this proposed model.

3. It is not clear about how to select in-context learning examples. Are they randomly selected or based on certain criteria?

4. Compared to the most relevant baseline MAE-VQGAN, the main contribution seems to be adding additional captions can provide guidance during model inference (as shown in Table 4). This observation is straightforward that may make the argument of in-context learning weak.

5. While authors provide analysis about S2CV and CCVF, it is not clear about how these types of datasets can help in-context learning.

---

> ### Author Response · Authors · 2024-08-02
>
> Thank you for the review and suggestions. We address your comments below.
>
> **Q:** Unfair comparison in Table 3.
>
> **A:** We provide fair apples-to-apples comparison in Table whereas in Table 3, we merely report the best performing result for eachvisual prompting method. Specifically, in Table 4, we report the performance of MAE-VQGAN and IMProv trained on the same dataset. IMProv is trained on CCVF which contains the same images from CVF and their associated captions. The results in Table 4 demonstrate that adding text promptsimprove the model performance.
>
> **Q:** While authors show that providing examples is beneficial to the model inference, the vision encoder takes a fixed image size. This may lead to a degraded performance when increasing the number of visual prompts. Such issue may limit the practical usage of this proposed model.
>
> **A:** We acknowledge that the current model could only take a fixed image size, as many other pre-train image encoders. We could only afford training with 224x224 resolution with academia level computation resources. However, if more computation budgets are available, this issue could be mitigated with higher training resolution.
>
> **Q:** It is not clear about how to select in-context learning examples. Are they randomly selected or based on certain criteria?
>
> **A:** We study the effect of choosing in-context learning examples based on different criteria in Figure 5, Figure 11 and Table 9. We conclude that usingIMProv with detailed text prompts leads to more robust downstream performance, thereby reducing the need to select specific in-context examples.
>
> **Q:** Compared to the most relevant baseline MAE-VQGAN, the main contribution seems to be adding additional captions can provide guidance during model inference (as shown in Table 4). This observation is straightforward that may make the argument of in-context learning weak.
>
> **A:** We would like to argue that by leveraging text guidance during inference, our model addressed the limitation of previous visual prompting methods. As Figure 4, 6 and 9 show, IMProv could solve ambiguity and provide more control.
>
> **Q:** While authors provide analysis about S2CV and CCVF, it is not clear about how these types of datasets can help in-context learning.
>
> **A:** As Figure 10 shows, our datasets cover diverse computer vision tasks, and some of the figures are in grid style. By training to reconstruct masked areas on these figures, our model may learn to predict the task completions by leveraging in-context examples in the grid.
>
> **Q:** The collected dataset can contribute to the multimodal understanding community if it can be open sourced.
>
> **A:** We promise to make our datasets publicly available upon acceptance.

---

### Review · Reviewer_47rH · 2024-07-18

**Summary Of Contributions:**

This paper is a multimodal version of Visual Prompting via Image Inpainting, which introduces a new multimodal prompted in-painting based computer vision, self supervised, in context learning framework. The prompt comes from two modalities: 1) visual prompt is coming from unmasked patches of the image 2) text prompt is coming from annotations (task, location, class names); The authors collect a new dataset from semantic scholar, and train a MAE-VQGAN model from scatch for this purpose. After training on collected cv paper images and laion images, the model can emerge some capabilities for various cv tasks.

**Audience:**

Yes

**Broader Impact Concerns:**

No ethical concerns .

**Claims And Evidence:**

Yes

**Requested Changes:**

1) Better discuss the difference and relation with Visual Prompting via Image Inpainting
2) Try more annotated supervised data like (SAM/depth anything/Grounded DINO/hed, etc) for Supervised ICL, I don't think instructpix2pix are suitable for depth/seg/hed/norm
3) Is it possible to also fine-tune SD inpainting model with your data recipe. It would show the benefit using MAE-VQGAN
4) Can authors provide some results like finetuning baseline in Table 6 with K=1,4,16 for IMProv, to see if it can scale to more shots

**Strengths And Weaknesses:**

Strengths:
1) It's good to see that using only in the wild laion data and cv paper data and training from scratch instead of initialized from pre-trained generative models, the model can achieve good results
2) Using VQ-GAN for generation offers better alignment and controlabiility than diffusion model as shown in figure 6 and 7
3) Text prompt helps to solve some hard cases in MAE-VQGAN as shown in Figure 9

Weaknesses:
1) Like previous work, this type of methods are trained on one-shot (2*2 grids), which is hard to extend to long context in-context learning naturally, and the num of shots are also barriered by resolution.

Summary:
The self supervised model's performance is still far from supervised ICL baseline. I acknowledge the contribution of this paper is good, but there is still a far way to achieve language model level in-context learning capability using in the wild data. I support this paper to be accepted, the contribution is good and the exploration is necessary. I appreciate the solid experiments in this paper. I hope the authors to make some changes and provide more experiments results, since it will provide benifitial insights into future research on this unsupervised vision in context learning area.

---

> ### Author Response · Authors · 2024-08-02
>
> Thank you for the review and suggestions. We address your comments below.
>
> **Q:** Like previous work, this type of methods are trained on one-shot (2*2 grids), which is hard to extend to long context in-context learning naturally, and the num of shots are also barriered by resolution.
>
> **A:** IMProv is not trained is  trained on 2x2 grids. Different from previous works (e.g, Painter, SegGPT), IMProv is trained to reconstruct randomly masked regions in images from the large-scale S2CV dataset we collected. In IMProv (ours) 2x2 grids are only introduced at test time. In Table 8 and Figure 16, we show that IMProv could generalize to more in-context examples like 3x3 and 4x4 grids. To improve the performance of more shots, we could fine-tune the model on high-resolution images with short schedule like EMU2 (Sun et. al.).
>
> **Q:** Better discuss the difference and relation with Visual Prompting via Image Inpainting
>
> **A:** Compared to VisualPrompt, the contribution of IMProv is two-fold. First of all, we introduce multi-modal prompting ability and demonstrate that text prompts could improve the model performance. We also explore the tradeoff between text and image prompts, showing that the text prompt could solve ambiguity and provide more control (Figure 4, 6). Secondly, we collect a new large dataset S2CV for multimodal in-context learning. Different from other image-text datasets, S2CV focuses on computer vision tasks. And it’s 3 times larger than the CVF dataset proposed in VisualPrompt.
>
> **Q:** Try more annotated supervised data like for Supervised ICL.
>
> **A:** In this work, we follow PromptDiffusion (Wang et. al. 2023) to use InstructPix2Pix dataset as supervised training data. Adding more annotated data is definitely one of our most important future works.
>
> **Q:** Is it possible to also fine-tune SD inpainting model with your data recipe?
>
> **A:** We have tried to finetune SD with the same training dataset as IMProv, and it only yielded 19.2 mIoU. Under the same setting, IMProv achieves 42.6 mIoU, which outperforms SD by a large margin.
>
> **Q:** Can authors provide some results like finetuning baseline in Table 6 with K=1,4,16 for IMProv, to see if it can scale to more shots?
>
> **A:** We provide the results of more shots in Table 8 and Figure 16. In Grid 4x4 setting, 5-shot outperforms 1-shot by 16.14 mIOU. It demonstrates IMProv could generalize to more few shot examples without any fine-tuning.

---

> > ### Comment · Reviewer_47rH · 2024-08-02
> > **Thanks for the response**
> >
> > Thank you so much for the response. I believe the authors have addressed my concerns

---

> > ### Comment · Reviewer_47rH · 2024-08-02
> > **Response cont.**
> >
> > I noticed that in Table 8, for grid 4*4, 7-shot results in 33.11, still worse than grid 2*2 1-shot's result 42.68. Can authors comment on this? Do you think simply doing AnyRes can solve this?

---

> > > ### Author Response · Authors · 2024-08-02
> > >
> > > Thanks for your prompt reply.
> > >
> > > Since the input size is fixed to 224x224 during training and inference, grid 4x4 makes each shot's resolution smaller compared to grid 2x2. It would be a more fair comparison if we could make the shot of the same resolution. Combining high-resolution techniques like AnyRes into our inpainting architecture should help resolve this issue.

---

> > > > ### Comment · Reviewer_47rH · 2024-08-02
> > > >
> > > > Thank you for the reply!

---

### Review · Reviewer_Qrwx · 2024-07-19

**Summary Of Contributions:**

This paper presents a approach to solve unseen computer vision tasks at test time. The contributions are two fold: the paper presents a new dataset of image-text pairs from Semantic Scholar; and the paper presents a method -- IMProv, to inpaint masked regions given
the rest of the image and a caption. The image here contains an in-context example in a 2x2 grid. The method is evaluated on various tasks such as foreground/background segmentation, edge and depth estimation.

**Audience:**

Yes

**Broader Impact Concerns:**

* The paper should include a Broader Impact Statement clarifying any ethical concerns regarding scraping data from Semantic Scholar.

**Claims And Evidence:**

Yes

**Requested Changes:**

* The paper should discuss it's novel aspects in more detail. In general, the introduction of the paper should be clearer including the "teaser" figure in Figure 1.
* The paper should expand Table 3 to include the evaluations discussed above.

**Strengths And Weaknesses:**

Strengths:
* The proposed dataset is interesting.
* The proposed approach shows promising results against prior work such as MAE-VQGAN, Stable Diffusion and classic 1-shot segmentation baselines. It also shows promising performance against prior visual prompting methods such as Painter.
* The paper is well written and easy to understand.
* The paper includes a variety of ablations.

Weakness:
* Novelty: The key differences of the proposed method IMProv to Painter (Wang et. al. 2023) is not clear. Painter also uses in-painting on a 2x2 grid. Both models are trained using masked image modelling.

* Do the results in Table 3 use the nearest neighbor in-context examples? What happens when an appropriate nearest neighbor  in-context example cannot be found? It would be better to report the results both with random in-context and nearest neighbor in-context examples.

* Comparison to Painter: In Table 3 it would be helpful to compare to Painter (Wang et. al. 2023), as the proposed method and Painter are both very similar.

* Comparison to supervised SOTA: Table 3 should also include the best supervised SOTA approaches, as results in Fig. 3 confirm that the proposed approach displays very weak performance on a variety of tasks. This will confirm is the proposed approach is usable for real-world tasks,

* Image to depth color bleeding: From Fig. 3 it appears that e.g. depth maps are incorrectly predicted -- are are color pixels instead of grayscale. How are such invalid cases handled?

* Can the proposed approach be extended to (multi-class) semantic segmentation or is it constrained to binary back/foreground segmentation?

---

> ### Author Response · Authors · 2024-08-02
>
> Thank you for the review and suggestions. We address your comments below.
>
> **Q:** The key differences between IMProv and Painter.
>
> **A:** We would like to emphasize that there are 2 major differences between IMProv and Pinater.
>
> Firstly, the architecture of Painter assumes 2x2 grid structure with exactly 1 in-context example. Our architecture is more general and flexible enough to be applied to an arbitrary number of in-context examples as shown in Figure 4 (no in-context examples) and Figure 16 (7 in-context examples).
>
> Secondly, Painter is trained with ground-truth segmentation annotations in a supervised manner, while our IMProv is trained on unlabeled image-text pairs collected from computer vision papers in a self-supervised manner, which doesn’t require ground-truth semantic segmentation annotation. Moreover, in Figure 13, we demonstrate our method could generalize better to other computer vision tasks like colorization, while Painter fails to do so.
>
> We also report the quantitative comparison of Painter, and our IMProv in the table below:
>
> | Model         | FSS-1000 mIoU |
> |---------------|---------------|
> | VisualPrompt  | 58.3          |
> | Painter       | 62.3          |
> | IMProv (CCVF) | 62.8          |
> | IMProv (S2CV) | 68.9          |
>
> Following Table 4 of Painter, we report the segmentation results on the FSS-1000 dataset. Our model outperforms Painter by a large margin. It’s also worth noting that changing the training dataset from CCVF to S2CV improves the mIoU significantly, which further indicates the effectiveness of our proposed S2CV dataset. We also added this table in the Table 11 of our revision.
>
> **Q:** Do the results in Table 3 use the nearest neighbor in-context examples? It would be better to report the results both with random in-context and nearest neighbor in-context examples.
>
> **A:** Table 3 follows the setting of VisualPrompt (Bar et. al. 2022), where we use random images from the same class as the in-context examples. We also report the results of different settings of in-context examples in Figure 5, Figure 11 and Table 9. It demonstrates that our IMProv with text prompt is more robust to the selection of in-context examples.
>
> **Q:** Comparison to supervised SOTA.
>
> **A:** In Table 3, we mainly compare with self-supervised approaches without training on ground-truth segmentation masks. Additionally, in Table 6, we compare with the fine-tuning baselines on Pascal VOC few shot segmentation tasks. Our method closes over 40% of the accuracy gap between MAE-VQGAN to supervised one-shot approaches. We also compared to SOTA methods like Painter in the Table above.
>
> **Q:** Image to depth color bleeding. How are such invalid cases handled?
>
> **A:** Our model is only trained on RGB images without depth map supervision, the depth is also predicted in RGB space. So the depth estimation results are not accurate enough. It could be further improved if we train the model with explicit depth map annotations.
>
> **Q:** Can the proposed approach be extended to (multi-class) semantic segmentation or is it constrained to binary back/foreground segmentation?
>
> **A:** Yes, our model can be generalized to multi-class semantic segmentation without fine-tuning. We included an example in Figure 3 (Image -> Segmentation section).
>
> **Q:** The paper should include a Broader Impact Statement clarifying any ethical concerns regarding scraping data from Semantic Scholar.
>
> **A:** We added the Broader Impact statement in the appendix of our revision.

---

### Decision · Action_Editor_vhXq · 2024-08-29

**Recommendation:** Accept with minor revision

**Comment:**

The reviewers appreciated the proposed dataset (Qrwx, 47rH), the achieved results (Qrwx, 47rH), the accompanying analyses (Qrwx, 47rH, UUw9), and the presentation (Qrwx, UUw9).

The main concerns raised by the reviewers were on the experimental comparisons, deemed potentially unfair due to the different pertaining datasets (UUw9), and the novelty and comparison with Painter (Wang et al., 2023, Qrwx), as well as concerns on the resolution and grid size (47rH, UUw9). The authors' response addressed most of the concerns, providing a comparison with Painter (now Table 11 of the revised manuscript), analyses on the grid size (Table 8), and clarifications on the comparisons.

Two reviewers recommended acceptance of the manuscript, valuing the effectiveness of the approach and the experimental efforts (47rH, UUw9), while Qrwx shared concerns regarding the technical novelty, the issues on resolutions, and the fact that the dataset is deemed the core contribution.

The AE finds that IMProv, while extending previous works, is still a valid contribution to the community (together with the proposed dataset) as its simplicity is an added value given that the topic is under-explored and complex baselines are not strictly necessary at this early stage. Thus, the AE recommends the acceptance of the work. In the final version, the authors are advised to perform some minor changes, i.e.:
- Given that the paper already discusses Painter and compares qualitatively IMProv with it (Fig. 8), Tab. 11 and the corresponding paragraph should be added to the main manuscript (fixing also the issue with the reference to the corresponding table).
- Include a discussion on the impact of the resolution and grid sizes within the main manuscript (following the discussion with 47rH and the limitation stressed in the reply to UUw9).

**Audience:**

In-context learning is a widespread topic in natural language processing still with little exploration in the context of visual applications, especially via multimodal inputs. This paper contributes to this direction and can constitute a reference for researchers in this emergent area of machine learning.

**Claims And Evidence:**

This work proposes IMProv, a method that performs in-context learning for visual applications. Specifically, IMProv focuses on the multimodal case, i.e., where the input is both visual and textual. Following previous ideas in this field, such as Painter (Wang et al., 2023) and Visual Prompting (Bar et al., 2022), IMProv treats this problem as an inpainting one, where the multimodal inputs are used to perform the inpainting task. To achieve this, the paper also collects a dataset, the Semantic Scholar Computer Vision dataset (S2CV) that contains figures from computer vision papers together with their captions.

The basic idea (and claim) of the paper is that (i) using text as additional conditioning for in-context learning can help to disambiguate difficult cases and better perform the target visual task; (ii) scaling data can further help to improve the performance. These claims are verified in multiple parts of the experimental section quantitatively (e.g., Tables 4 and 6 for both text conditioning and scale, Table 5 for scale, margins in Table 3 for their union, Fig. 5 for the text only) and with accompanying qualitative examples (e.g., analyses of Fig. 8 and Fig. 9 on failure cases and comparisons with Painter).